# Synthesis and Crystal Structure of Ilmenite-Type Silicate with Pyrope Composition

**Takayuki Ishii [1],\* , Ryosuke Sinmyo [2] and Tomoo Katsura [3]**

[1] Institute for Planetary Materials, Okayama University, Misasa 682-0193, Japan

[2] Department of Physics, School of Science and Technology, Meiji University, 1-1-1 Higashi Mita, Tama-ku, Kawasaki 214-8571, Japan; sinmyo@meiji.ac.jp

[3] Bavarian Research Institute of Experimental Geochemistry and Geophysics, University of Bayreuth, 95440 Bayreuth, Germany; tomo.katsura@uni-bayreuth.de

\* Correspondence: takayuki.ishii@okayama-u.ac.jp

**Abstract:** Akimotoite, ilmenite-type $MgSiO_3$ high-pressure polymorph can be stable in the lower-mantle transition zone along average mantle and subducting slab geotherms. Significant amounts of $Al_2O_3$ can be incorporated into the structure, having the pyrope ($Mg_3Al_2Si_3O_{12}$) composition. Previous studies have investigated the effect of $Al_2O_3$ on its crystal structure at nearly endmember compositions. In this study, we synthesized high-quality ilmenite-type $Mg_3Al_2Si_3O_{12}$ phase at 27 GPa and 1073 K by means of a Kawai-type multi-anvil press and refined the crystal structure at ambient conditions using a synchrotron X-ray diffraction data via the Rietveld method to examine the effect of $Al_2O_3$. The unit-cell lattice parameters were determined to be $a = 4.7553(7)$ Å, $c = 13.310(2)$ Å, and $V = 260.66(6)$ Å$^3$, with $Z = 6$ (hexagonal, $R\bar{3}$). The volume of the present phase was placed on the akimotoite-corundum endmember join. However, the refined structure showed a strong nonlinear behavior of the $a$- and $c$-axes, which can be explained by Al incorporation into the $MgO_6$ and $SiO_6$ octahedral sites, which are distinctly different each other. Ilmenite-type $Mg_3Al_2Si_3O_{12}$ phase may be found in shocked meteorites and can be a good indicator for shock conditions at relatively low temperatures of 1027–1127 K.

**Keywords:** ilmenite; akimotoite; pyrope; high pressure; X-ray diffraction; crystal structure; Rietveld analysis; mantle; subducting slab; corundum

## 1. Introduction

$Mg_3Al_2Si_3O_{12}$ is a major component in mantle garnet, called pyrope (Prp). Prp is stable up to uppermost-lower-mantle conditions and transforms to bridgmanite and corundum (post-garnet phase) along an average mantle geotherm [1–5]. Along a cold subducting slab geotherm, Prp may first transform into an ilmenite-type phase and then into the post-garnet phase at mantle transition zone pressures [6]. The ilmenite-type $Mg_3Al_2Si_3O_{12}$ may play an important role in understanding mantle structure and dynamics [4].

Akimotoite (Aki) is a high-pressure polymorph of $MgSiO_3$ minerals with the ilmenite structure belonging to the $R\bar{3}$ space group (e.g., [7]). It is stabilized at approximately 18–25 GPa and 1000–2000 K (e.g., [8–11]). The Aki structure consists of distorted, hexagonal, close-packed oxygen anions and magnesium and silicon cations occupying octahedral interstitials. Each of the edge-shared $MgO_6$ and $SiO_6$ octahedra forms distinct layers, which are connected to each other by face-sharing. The ideal Aki structure has Mg and Si completely ordered in alternating layers along the $c$-axis. The $MgO_6$ octahedra exhibit a significant degree of distortion in comparison with the $SiO_6$ octahedra due to a large displacement of the $Mg^{2+}$ cation from the center of the octahedron.

Aki can form a solid solution with $Al_2O_3$ component by the coupled substitution of $Mg^{2+}$ + $Si^{4+}$ to $2Al^{3+}$ up to 1–2 mol% and ~25 mol% at relatively high temperatures of 1700–1873 K and 1073–1173 K, respectively (e.g., [1,6,12–15]). The $Al_2O_3$ endmember in

the Aki binary solid solution is generally assumed to be corundum (Crn) with the space group of $R\bar{3}c$ because of its structural similarity to Aki. The Crn structure consists of $AlO_6$ face-sharing octahedral layers along the *c*-axis, in which all the cation sites are equivalent, resulting in a more symmetric structure than the Aki structure. A Raman spectroscopy study showed that the space group of Aki with 25 mol% $Al_2O_3$, corresponding to the Prp composition (Prp-Aki) is not $R\bar{3}c$, but $R\bar{3}$ [16]. This structural change from the $Al_2O_3$ endmember may hamper the formation of a complete solid solution in the $MgSiO_3$-$Al_2O_3$ system and result in a strong nonlinear change in lattice parameters in the Aki structure with increasing $Al_2O_3$ content [6,15,17,18]. We note that the transition alumina content is still unclear.

Previous-phase relation studies have reported that Aki with relatively low $Al_2O_3$ content exists in pyrolite and harzburgite lithologies at mantle transition zone pressures (e.g., [9,19–22]), which might explain the relatively high seismic velocities and anisotropy at this depth (e.g., [11,23–31]). As mentioned above, Aki can accommodate up to 25 mol% $Al_2O_3$ at relatively low temperatures of cold subducting slabs. Thus, Prp may transform to the ilmenite-type phase and persist down to uppermost-lower-mantle conditions [6].

Additionally, natural Aki with $(Mg,Fe)SiO_3$ composition has been discovered in meteorites experiencing high pressure–temperature conditions, such as the Tenham chondritic meteorite (e.g., [32]). The knowledge of structure and stability fields of high-pressure minerals in meteorites is useful for the constraint of conditions of shock events (e.g., [33–38]). Therefore, Aki is also an important mineral for understanding the origin of shocked meteorites.

Despite its significance, no structural refinement of Prp-Aki has ever been conducted, and therefore, the knowledge of structural changes in Aki due to the significant incorporation of $Al_2O_3$ components is limited. In this study, we synthesize high-quality Prp-Aki using a Kawai-type multi-anvil press and report its crystal structure obtained by Rietveld analysis [39].

## 2. Materials and Methods

A starting material for the synthesis of Prp-Aki was prepared from a mixture of MgO, $Al_2O_3$, and $SiO_2$ with a molar ratio of 3:1:3. The mixture was heated at 1950 K and kept for 1 h, at which point the sample was completely melted. After maintaining the temperature, the sample was quenched in water to produce a glass with Prp composition (Prp-glass). Py-Aki was synthesized from the Prp-glass powder using the Kawai-type multi-anvil press with an Osugi-type guide block, IRIS-15, at the Bayerisches Geoinstitut, University of Bayreuth (BGI) [12,40]. Pressure calibration was reported in [12]. Tungsten carbide anvils (Fujilloy, TF05) with truncated edge lengths of 3.0 mm were used in combination with a 5 wt% $Cr_2O_3$-doped MgO pressure medium with an edge length of 7 mm. The cell assembly was essentially the same as that used in [12]. A cylindrical Mo furnace was placed in the center of the pressure medium. The powdered starting material of Prp-glass was placed at the central part of the furnace. A $ZrO_2$ sleeve and end plugs were placed outside of the Mo furnace and at both ends of the furnace, respectively, for thermal insulation. Temperature was measured at the central part of the outer surface of the furnace using a W-3%Re/W-25%Re thermocouple. No correction was made for the effect of pressure on the electromotive force of the thermocouple. The sample was compressed to the target pressure of 27 GPa for 4 h, and then the temperature was increased to 1073 K by applying electrical power at a rate of ~100 K/min and kept in the range of $\pm$5 K. After maintaining the target temperatures for 1 h, the electrical power was turned off to quench the sample, and then gradual decompression was carried out for 12 h.

The recovered sample with a cylinder shape (a diameter of ~0.5 mm and height of ~1 mm) was identified using a micro-focus X-ray diffractometer (D8 DISCOVER, Bruker Corporation, Billercia, MA, USA) with a 2D VÅNTEC500 solid-state detector. X-ray diffraction data were collected using 50 μm Co-K$\alpha$ radiation focused with a polycapillary mini-lens operating at 40 kV and 500 mA. The composition of the sample was analyzed using an electron microprobe analyzer with a wavelength-dispersive spectrometer (JXA-8200, JEOL

LtD., Tokyo, Japan) operating at an accelerated voltage of 15 kV and a probe current of 15 nA.

An angle-dispersive synchrotron powder X-ray diffraction pattern for the Rietveld analysis was collected in the BL10XU at SPring-8, Hyogo, Japan [41]. The collimated X-ray beam, 10–20 μm in diameter and monochromatized by a Si double monochromator, was irradiated onto the crushed sample fixed in a rhenium foil ring. The two-dimensional (2D) diffraction patterns were obtained with a charge-coupled device (APEX, Bruker Corporation, Billercia, MA, USA) for 1 s while the sample was rotated. A diffraction pattern of $CeO_2$ standard was used to calibrate the X-ray wavelength ($\lambda = 0.41429$ Å) and camera length. The 2D profile of the sample was converted to the one-dimensional (1D) diffraction profile using IPAnalyzer software (ver3.956) [42].

Rietveld analysis was conducted with the RIETAN-FP/VENUS package [43]. The structural parameters of $MgSiO_3$ Aki determined by single crystal X-ray diffraction were used as the initial structure model [7]. The lattice parameters determined from the 1D X-ray diffraction pattern were used as the initial values in the Rietveld refinement. $SiO_2$ stishovite with a grain size of ~1 μm was found and included as the secondary phase in the Rietveld analysis. Because the electron microprobe analysis of the recovered Prp-Aki with a grain size of 1–3 μm, it was shown that the composition was $Mg_{2.99(2)}Al_{2.01(2)}Si_{3.01(1)}O_{12}$, the chemical composition was set to $Mg_3Al_2Si_3O_{12}$, and the site occupancies of two octahedral sites were fixed to $(Mg_{0.75}Al_{0.25})O_6$ and $(Si_{0.75}Al_{0.25})O_6$, assuming that Mg and Si atoms were completely ordered as reported in $MgSiO_3$ Aki [7], but Al atoms occupied the two sites equally. The final reliability values ($R_{wp}$, $R_e$, $R_B$, and $R_F$) obtained from the above constraints were less than 5% (Table 1).

**Table 1.** Structural parameters of ilmenite-type $Mg_{2.99(2)}Al_{2.01(2)}Si_{3.01(1)}O_{12}$.

| Atom | Wyckoff Site | $g$(Mg or Si) | $g$(Al) | $x$ | $y$ | $z$ | $U_{iso}$ (Å$^2$) |
|---|---|---|---|---|---|---|---|
| M1 [1] | $6c$ | 0.75 [2] | 0.25 [2] | 0 | 0 | 0.1410(2) | 0.0161(6) |
| M2 [1] | $6c$ | 0.75 [2] | 0.25 [2] | 0 | 0 | 0.3442(1) | 0.0164(7) |
| O | $18f$ | - | - | 0.3664(4) | 0.0097(7) | 0.0788(1) | 0.0176(5) |

The crystal system, space group, and lattice parameters: Hexagonal, $R\bar{3}$, $a = 4.7553(7)$ Å, $c = 13.310(2)$ Å, $V = 260.66(6)$ Å$^3$, $Z = 6$, $V_m = 26.161(6)$ cm$^3$/mol, and $D = 3.852(1)$ g/cm$^3$. The reliability indexes: $R_{wp} = 3.574\%$, and $R_e = 3.170\%$. Aki phase: $R_B = 1.751\%$, $R_F = 0.839\%$. $SiO_2$ stishovite: $R^B = 1.113\%$, $R_F = 0.571\%$ (the estimated amount: ~1 wt.%). $R_{wp} = \left\{ \frac{\sum_i w_i [y_i - f_i(x)]^2}{\sum_i w_i y_i^2} \right\}^{1/2}$, $R_B = \frac{\sum_K |I_0(h_K) - I(h_K)|}{\sum_K I_0(h_K)}$, $R_F = \frac{\sum_K ||F_0(h_K)| - |F(h_K)||}{\sum_K |F_0(h_K)|}$, $R_e = \left\{ \frac{N-P}{\sum_i w_i y_i^2} \right\}^{1/2}$, where $y_i$, $w_i$ and $f_i(x)$ are the intensity observed at step i, the statistical weight, and the theory intensity, respectively. $I_0(h_K)$, $I(h_K)$, $F_0(h_K)$, and $F(h_K)$ are the observed and calculated intensities and structure factors for the reflection K, respectively. $N$ and $P$ are the numbers of data points and refined parameters, respectively. $g$(M): site occupancy of M. [1] M1 and M2 atomic positions are occupied by Mg or Al and Si or Al, respectively. [2] Site occupancies are fixed.

## 3. Results

Figure 1 shows the result of Rietveld fitting of the present sample. Lattice parameters of $a = 4.7553(7)$ Å, $c = 13.310(2)$ Å, and, $V = 260.66(6)$ Å$^3$ were obtained after the refinement (Table 1). These lattice parameters are consistent with those determined by [6]. The $a$ and $c$ axes of Prp-Aki were larger and smaller, respectively, than those of $MgSiO_3$ Aki ($a = 4.7284$ Å and $c = 13.5591$ Å) [7]. As a result, the obtained volume was smaller than that of $MgSiO_3$ Aki ($V = 262.54$ Å$^3$). The $c/a$ ratio decreased with $Al_2O_3$ content in the Aki-Crn join ($MgSiO_3$ Aki: 2.87, Prp-Aki: 2.80, and $Al_2O_3$ Crn: 2.73) [6]. The $c/a$ ratio in this study was 2.80, which is consistent with the trend and the values previously obtained in Prp-Aki.

Figure 2 shows the crystal structure of Prp-Aki after the structure refinement. Table 2 summarizes the interatomic distances and angles. The bond valence sums (BVS) [44] and effective coordination numbers ($n_c$) [45] were calculated from the interatomic distances of Prp-Aki. The refined structure had two types of oxygen-octahedral sites: $M1O_6$ (M1 = $Mg_{0.75}Al_{0.25}$) and $M2O_6$ (M2 = $Si_{0.75}Al_{0.25}$). An $M1O_6$ octahedron was connected with an edge of a neighboring

M1O$_6$ octahedron in the *ab* plane. Similarly, an M2O$_6$ octahedron (M2 = Si$_{0.75}$Al$_{0.25}$) was connected with an edge of a neighboring M2O$_6$ octahedron. An M1O$_6$ octahedral layer and an M2O$_6$ octahedral layer alternated along the *c*-axis. M1O$_6$ and M2O$_6$ octahedra were connected by face-sharing and point-sharing with M2O$_6$ octahedra. The average cation–oxygen distances in M1O$_6$ and M2O$_6$ octahedra were 2.014 Å and 1.854 Å, respectively. The ionic radii of Mg$^{2+}$, Al$^{3+}$, Si$^{4+}$, and O$^{2-}$ with six-fold coordination were $r_{Mg}$ = 0.72, $r_{Al}$ = 0.535, $r_{Si}$ = 0.4, and $r_O$ = 1.40 Å [46]. The ideal M1-O and M2-O distances for an ionic structure calculated with the cation radii were 0.75 $r_{Mg}$ + 0.25 $r_{Al}$ + $r_O$ (2.073 Å) and 0.75 $r_{Si}$ + 0.25 $r_{Al}$ + $r_O$ (1.833 Å), respectively. The observed average interatomic distances were similar to the calculated ones and were significantly smaller and larger than the Mg-O and Si-O distances of MgSiO$_3$ Aki, respectively. The BVS values in the M1O$_6$ and M2O$_6$ octahedra were also higher and lower than +2 and +4 (+2.55 and +3.31, respectively). The $n_c$ values of M1 (5.31) and M2 (5.82) suggest six-coordination of each cation by oxygen. The results suggest the incorporation of Al into each site.

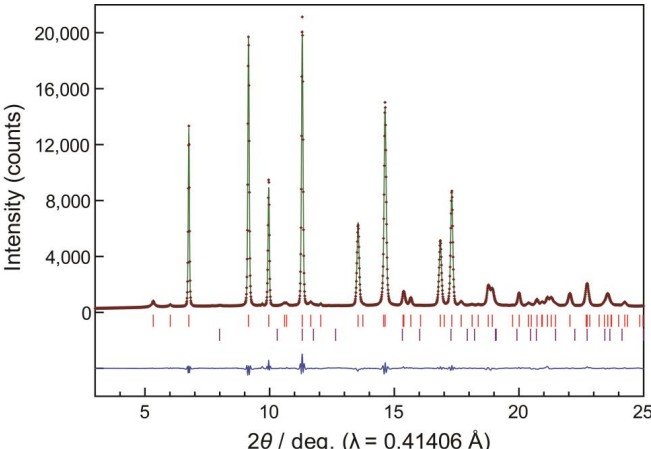

**Figure 1.** The result of the Rietveld refinement of ilmenite-type Mg$_3$Al$_2$Si$_3$O$_{12}$. The synchrotron X-ray diffraction pattern at atmospheric pressure and room temperature was used for structure refinement. The observed data and the calculated profiles are shown by dots (brown) and solid (green) lines, respectively. The intensity difference between the observed (brown) and calculated (green) patterns is shown at the bottom (blue). The Bragg peak positions for ilmenite-type Mg$_3$Al$_2$Si$_3$O$_{12}$ and rutile-type SiO$_2$ (stishovite) are indicated by upper (red) and lower (purple) ticks, respectively.

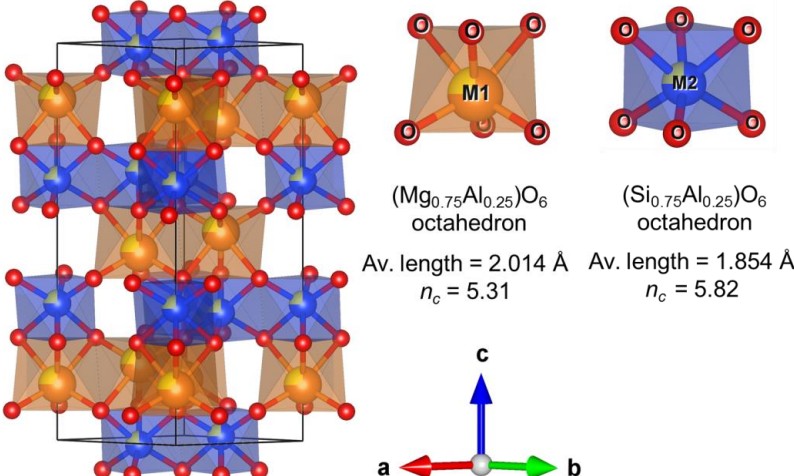

**Figure 2.** Crystal structure of refined ilmenite-type Mg$_3$Al$_2$Si$_3$O$_{12}$. The solid line in the structure shows the unit cell. The occupancies of Mg, Al, and Si in the M1 and M2 sites are represented by the sub-divided areas in each circle. The crystal structure was drawn using VESTA software (version 3) [47].

**Table 2.** Interatomic distances and angles in the structures of ilmenite-type $Mg_{2.99(2)}Al_{2.01(2)}Si_{3.01(1)}O_{12}$.

**Bond Length (Å)**

| | | | |
|---|---|---|---|
| M1−O $^i$ × 3 | 1.909(3) | M2−O $^i$ × 3 | 1.802(2) |
| M1−O $^{ii}$ × 3 | 2.120(2) | M2−O $^{iii}$ × 3 | 1.906(2) |
| Average | 2.014 | Average | 1.854 |
| BVS | 2.55 | BVS | 3.31 |
| $n_c$ | 5.31 | $n_c$ | 5.82 |

**Bond Angles (°)**

| | | | | | |
|---|---|---|---|---|---|
| O $^i$−M1−O | 102.6(1) | O $^{ii}$−M1−O $^{vi}$ | 74.79(8) | O−M1−O $^{iv}$ | 160.1(1) |
| O $^v$−M2−O $^{iii}$ | 97.00(8) | O $^{ii}$−M2−O $^{iii}$ | 84.96(9) | O $^{vii}$−M2−O $^{viii}$ | 168.8(1) |
| M1−O $^{viii}$−M1 $^{ii}$ | 89.07(9) | M2 $^{ix}$−O $^x$−M2 $^{xi}$ | 96.2(1) | M1−O $^{iii}$−M2 | 84.23(6) |
| M1−O $^{vi}$−M2 $^{xii}$ | 137.5(2) | | | | |
| M1−O$^i$ × 3 | 1.909(3) | M2−O $^i$ × 3 | 1.802(2) | | |

Symmetry codes: (i) $-y$, $x-y$, $z$. (ii) $-x + 2/3$, $-y + 1/3$, $-z + 1/3$. (iii) $x + 2/3$, $y + 1/3$, $z + 1/3$. (iv) $x + 2/3$, $-x + y + 1/3$, $-z + 1/3$. (v) $-y + 2/3$, $x-y + 1/3$, $z + 1/3$. (vi) $x-y + 2/3$, $y + 1/3$, $-z + 1/3$. (vii) $-x + y+2/3$, $-y + 1/3$, $z + 1/3$. (viii) $y +2/3$, $-x + y+1/3$, $-z + 1/3$. (ix) $-x$, $-y$, $-z$. (x) $-x + y+1/3$, $-x +2/3$, $z + 2/3$. (xi) $x + 2/3$, $y + 1/3$, $z + 1/3$. (xii) $-x + 1/3$, $-y + 2/3$, $-z + 2/3$. BVS: bond valence sum value. $n_c$: effective coordination number.

## 4. Discussion

### 4.1. Nonlinear Structural Feature of Prp-Aki

As shown in Figure 3, the lattice parameters determined in this study are consistent with the trend in the Aki $MgSiO_3$-$Al_2O_3$ binary solid solution ([15] and references therein). This trend is reasonable because $Al_2O_3$ Crn has a longer *a*-axis and a shorter *c*-axis than $MgSiO_3$ Aki. In the Aki structure, the *a*- and *c*-axes generally increase and decrease, respectively, with $Al_2O_3$ content. These trends are strongly nonlinear when compared with the ideal endmember-join line. On the other hand, the volume of Aki changes almost linearly with the $Al_2O_3$ content. Figure 4 summarizes the structural change with increasing $Al_2O_3$ content. The volumes of $MgO_6$ and $SiO_6$ octahedra decrease and increase with $Al_2O_3$ content, respectively (Figure 4a). This is a reasonable trend with the replacement of larger ($Mg^{2+}$: 0.720 Å) and smaller ($Si^{4+}$: 0.400 Å) cations into $Al^{3+}$ (0.535 Å), respectively, although the refined values are not on the endmember join line, suggesting the nonlinear behavior. The average octahedral volume is on the endmember join line. Another feature of the Aki structure can support interpretation of the non-linear behavior. There are two oxygen–oxygen (O-O) bonds in the (110) plane and along the *c* axis, which are directly linked with the lattice parameters of *a* and *b* and *c*, respectively (Figure 4b,c). The O-O distance in the (110) plane is along the shared face and increases rapidly with increasing $Al_2O_3$ content compared with the endmember join line (Figure 4b). On the other hand, the O-O distance along the *c* axis corresponds to the height of the face-shared octahedra and is lower than that of the trend of the Aki-Crn endmember join (Figure 4c). These changes show the same trend as the lattice parameter change and should directly correspond to the lattice parameter change. Therefore, the resulting structural data can explain the lattice parameter change with $Al_2O_3$ content. In the Aki structure, Mg and Si are ordered in each site, whereas all cations are disordered in the Crn structure. Thus, these two structures are distinctively different. We note that the present refinement cannot provide information on disordering between the two sites because of similar X-ray scattering factors in these atoms. A future neutron diffraction study may be able to discuss possible disordering.

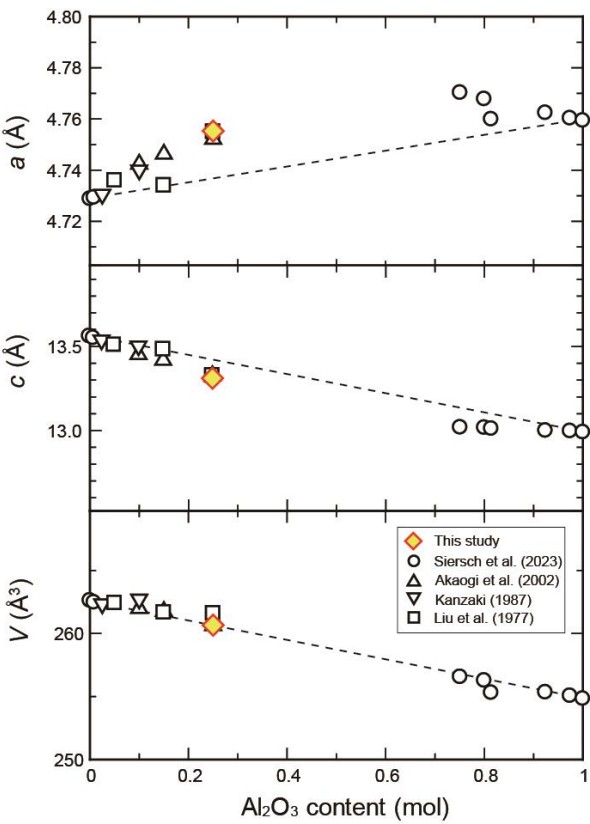

**Figure 3.** Lattice parameter changes in the MgSiO$_3$ akimotoite-Al$_2$O$_3$ corundum system. The dash lines show the endmember join between Aki and Crn [6,15,17,18].

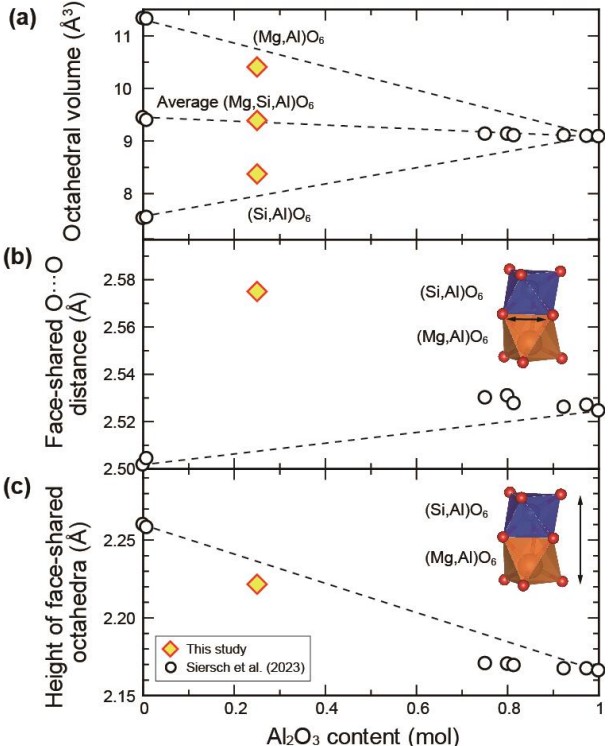

**Figure 4.** Structural change in the MgSiO$_3$ akimotoite-Al$_2$O$_3$ corundum system. (**a**) Octahedral volume. (**b**) Face-shared O-O distance in the (110) plane. (**c**) Average height of two face-shared octahedra along the *c* axis [15]. The dashed lines show the endmember join between Aki and Crn. The octahedra in (**b**,**c**) were drawn using VESTA software (version 3) [47].

### 4.2. Comparison with Compounds in the Mg₃Al₂Si₃O₁₂ System

Five polymorphs have been discovered in the $Mg_3Al_2Si_3O_{12}$ system: Prp, orthorhombic perovskite-type phase (Prp-Bdm), lithium niobate-type (Prp-LN) phase, and postBdm-type phase (Prp-pBdm), in addition to Prp-Aki presented in this study (e.g., [1,6,13,48]) (Figure 5). Pyp has three non-equivalent sites, in which Mg, Al, and Si are accommodated in dodecahedral, octahedral, and tetrahedral sites, respectively, resulting in the lowest density of five polymorphs ([13] and references therein). In the Prp-Bdm phase, Mg and Si are accommodated in bicapped trigonal prism and octahedral sites, respectively. Al is assumed to be equally accommodated into these two sites [1,3]. We note that structure analysis has not been performed. Prp-pBdm with $CaIrO_3$-type orthorhombic structure is the densest phase of the five phases, which forms from Prp-Bdm above 130 GPa [49–51]. Although its structure analysis has also not been conducted, this phase may consist of corner- and edge-shared $(Si_{0.75},Al_{0.25})O_6$ layers and bicapped trigonal prism $(Mg_{0.75},Al_{0.25})O_8$. The Py-LN phase has hexagonal symmetry with the space group of $R3c$ and two octahedral sites for Mg and Si, in which Al is also considered to be equally incorporated into both sites because of back-transformation from the Prp-Bdm phase during cold decompression [13]. This phase is formed by tilting of octahedra due to the decrease in ionic radius ratio between the A-site cation and oxygen during decompression. Thus, the Prp-LN phase has a layer with alternating edge-sharing $AO_6$ and $BO_6$ octahedra. Although the Prp-LN structure is quite similar to the Prp-Aki structure, alternating layers of edge-sharing $AO_6$ and $BO_6$ in Prp-Aki create a distinctive difference between the two structures. The density of Prp-Aki is approximately 2% lower than that of the LN phase [13]. This lower density would be a result of the longer O-O distance along the *c*-axis (Prp-LN: 2.13 Å and Prp-Aki: 2.22 Å) due to the presence of Mg-Si ordered layers in Prp-Aki ([6,13], this study).

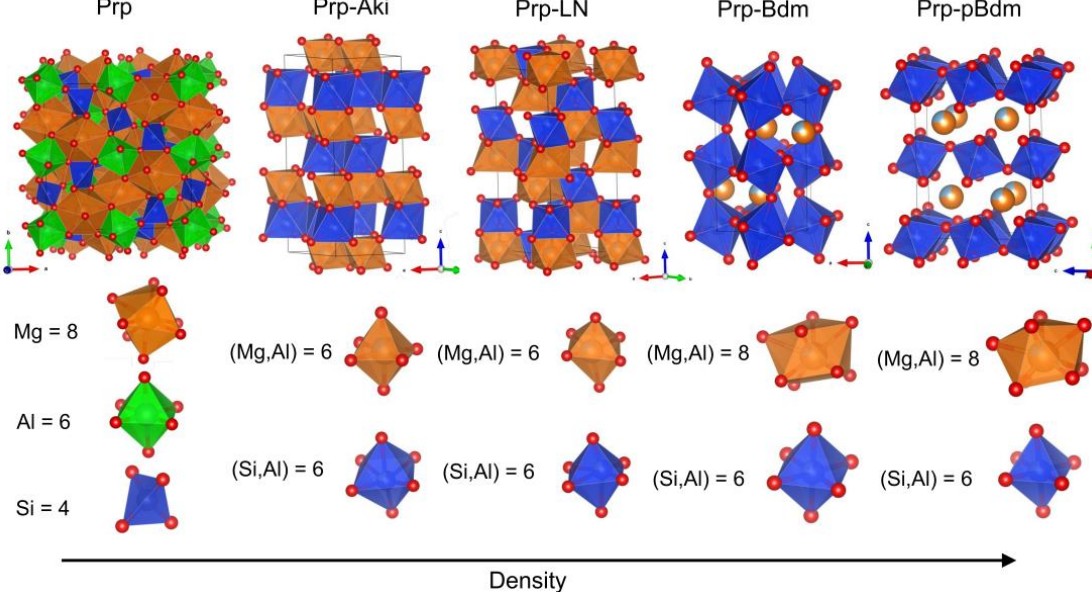

**Figure 5.** Crystal structures of polymorphs in the $Mg_3Al_2Si_3O_{12}$ system. Atomic distributions of Prp, Prp-Aki, Prp-LN, Prp-Bdm, and Prp-pBdm were taken from [52], this study, [13,49], and [53], respectively. The numbers in the bottom of the figure express coordination numbers of cations in each site. The crystal structures were drawn using VESTA software (version 3) [47].

### 4.3. Possible Formation of Prp-Aki in Shocked Meteorites and Possible Indicator of Pressure-Temperature Condition on Shock Events

Some natural ilmenite-type Al-bearing $(Mg,Fe)SiO_3$ has been discovered in shocked meteorites such as Tenham and Suizhou ([54] and references therein). Several types of Aki have been found so far. For example, Aki with 21–22 mol%$FeSiO_3$ coexisting with clinoenstatite was discovered in the shocked-metamorphosed Tenham chondrite [32]. Such

a high Fe-bearing Aki is unstable under equilibrium conditions, suggesting the direct transformation from pyroxene by a martensitic-type shear transformation mechanism due to the relatively short shock event on the order of $10^{-2}$ to $10^{-1}$ s. Another example is Aki, with 4.4 wt% $Al_2O_3$, coexisting with Bdm, ringwoodite, and silicate glass in an L6 shocked chondrite [55]. This Aki is considered to be metastably crystallized from melt during fast cooling. Similarly, Prp-Aki might be formed from melt after quenching in shocked meteorites of $Al_2O_3$-rich systems such as Prp systems. Furthermore, shock-induced minerals from the Xiuyan crater, which were considered to be exposed at relatively low temperatures (1073–1173 K) and 25–45 GPa, were also found [56]. If Prp-Aki were discovered in such a high-pressure and low-temperature shocked region, this phase would be a good indicator to estimate the shocked pressure and temperature conditions, because Prp-Aki is stable only at 25–27 GPa and 973–1273 K [6].

## 5. Conclusions

We synthesized ilmenite-type silicate with $Mg_3Al_2Si_3O_{12}$ composition at 27 GPa and 1073 K and analyzed its crystal structure using Rietveld analysis. The refinement showed that volume of the present phase is on the $MgSiO_3$-$Al_2O_3$ join, whereas the lattice parameters are highly anisotropic. Such nonlinear features can be interpreted by changes in oxygen–oxygen bonds with the alumina content. Our data clearly showed structural differences between the Prp-Aki and Prp-LN phases, both of which have similar but different octahedral arrangements, resulting in a lower-density phase of Prp-Aki. The formation of Prp-Aki can be expected in shocked meteorites, and can creative a narrow constraint of shock conditions of 25–27 GPa and 973–1273 K.

**Author Contributions:** Conceptualization, T.I.; methodology, T.I.; software, T.I.; validation, T.I.; formal analysis, T.I.; investigation, T.I., R.S.; resources, T.I., R.S. and T.K.; data curation, T.I.; writing—original draft preparation, T.I.; writing—review and editing, T.I., R.S. and T.K.; visualization, T.I.; supervision, T.I.; project administration, T.I. and T.K.; funding acquisition, T.K. All authors have read and agreed to the published version of the manuscript.

**Funding:** This research was funded by the research grants approved by the European Research Council (ERC) under the European Union's Horizon 2020 research and innovation programme (Proposal No. 787 527) to T.K. and by the Japan Society for the Promotion of Science (JSPS) KAKENHI Grants (Number 23K19067 and 24K00735) to T.I.

**Data Availability Statement:** All data generated or analyzed during this study are included in this published article.

**Acknowledgments:** We thank four anonymous reviewers for their constructive comments. The synchrotron XRD measurements were carried out at the BL10XU of SPring-8 with the approval of the Japan Synchrotron Radiation Research Institute (JASRI).

**Conflicts of Interest:** The authors declare no conflict of interest.

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
