# Peer review of "Synthesis and Crystal Structure of Ilmenite-Type Silicate with Pyrope Composition"

_solids, doi:10.3390/solids5030026_

Round 1

Reviewer 1 Report

Comments and Suggestions for Authors

A nice paper, good to see interesting science being done by synchrotron X-ray Powder Diffraction. I think that this paper will be suitable for publication after a few minor changes are made to the text. I suggest the following changes.

Lines 52-56. Do the authors know at what Al content does the space group change from R-3c to R-3? If a sample were to be made with a greater Al content at what composition would the space group change?

Line 73. Give reference to Rietveld paper from 1969.

Lines 96-97. What were the results of this XRD analysis, were lattice parameters refined and how did they compare with the synchrotron results?

Line 103. Is there a reference for the synchrotron instrument?

Lines 113-114. How much stishovite (SiO2) was present as a second phase? I see that the authors assumed 3:1 ratios of Mg:Al and Si:Al, the presence of a second phase may have changed the Si:Al ratio. Did the authors try to refine the M1 and M2 site occupancies?

Lines 114-119. Were any interatomic distance constraints used for the Mg/Al-O and Si/Al-O distances in the Rietveld refinement?

Author Response

Comment 1: A nice paper, good to see interesting science being done by synchrotron X-ray Powder Diffraction. I think that this paper will be suitable for publication after a few minor changes are made to the text. I suggest the following changes.

Response 1: Thank you for your review. We made responses to your comments below.

Comment 2: Lines 52-56. Do the authors know at what Al content does the space group change from R-3c to R-3? If a sample were to be made with a greater Al content at what composition would the space group change?

Response 2: The transition alumina content is unknown yet. It would be interesting to investigate the transition alumina content in future. We added the following sentence to explain current understanding about the transition alumina content in lines 56-57.

“We note that the transition alumina content is unclear yet.”

Comment 3: Line 73. Give reference to Rietveld paper from 1969.

Response 3: We cited the following reference here as the reviewer probably suggested.

Rietveld, H. M. (1969). A profile refinement method for nuclear and magnetic structures. Applied Crystallography2(2), 65-71.

Comment 4: Lines 96-97. What were the results of this XRD analysis, were lattice parameters refined and how did they compare with the synchrotron results?

Response 4: As described here, we used the micro-focus X-ray diffractometer only identify phases present because micro-focus XRD does not provide precise lattice parameters compared with a conventional powder X-ray diffractometer due to its geometrical issue.

Comment 5: Line 103. Is there a reference for the synchrotron instrument?

Response 5: We cited here two references about the synchrotron instrument as below.

“Hirao, N., Kawaguchi, S. I., Hirose, K., Shimizu, K., Ohtani, E., & Ohishi, Y. (2020). New developments in high-pressure X-ray diffraction beamline for diamond anvil cell at SPring-8. Matter and Radiation at Extremes5(1).”

“Kawaguchi-Imada, S., Sinmyo, R., Ohta, K., Kawaguchi, S., & Kobayashi, T. (2024). Submillisecond in situ X-ray diffraction measurement system with changing temperature and pressure using diamond anvil cells at BL10XU/SPring-8. Journal of Synchrotron Radiation31(2), 343-354.”

Comment 6: Lines 113-114. How much stishovite (SiO2) was present as a second phase? I see that the authors assumed 3:1 ratios of Mg:Al and Si:Al, the presence of a second phase may have changed the Si:Al ratio. Did the authors try to refine the M1 and M2 site occupancies?

Response 6: The present Rietveld analysis showed ~1 wt.% of stishovite. Ishii et al. (2017) checked the composition of pyrope glass starting material, which was Mg2.93(2)Al2.00(2)Si3.04(2)O12, indicting a slightly Si-rich starting sample. This would be a reason why we had a small amount of stishovite. Although the sample is in a silica-enriched system, we got a composition consistent with pyrope within an error of ~0.3%. Examination of cite occupancies in such a small error by XRD method is not realistic. More importantly, X-ray scattering factors of Mg, Si, Al are very similar each other. Therefore, it is almost impossible to examine cite occupancies among these atoms. Due to this reason, we fixed cite occupancies in this manuscript.

We added the amount of stishovite determined by the Rietveld analysis as below in the caption of Table 1 (line144).

“(the estimated amount: 1 wt.%)”

Comment 7: Lines 114-119. Were any interatomic distance constraints used for the Mg/Al-O and Si/Al-O distances in the Rietveld refinement?

Response 7: We have not used any constraints on interatomic distances. Comparing with the previous structural data of ilmenite-type and corundum type phases in the current system and structural features of other oxide minerals, the structural features of the present phase are reasonable. We therefore think that our refinement was reasonably converged and it is not necessary to constrain them.

Reviewer 2 Report

Comments and Suggestions for Authors

Authors firstly report crystal chemistry  of ilmentie-type silicate with pyroe composition, whose data quality are high. They also well explain the strong nonlinear behavior of the a- and c-axes with Al2O3 content in the MgSiO3-Mg3Al2Si3O12 system. These results are significant of high-pressure mineral physics, such as in deep earth and shocked meteorite. This manuscript is well organized.  Here I high recommend it in present form. 

Author Response

Comment: Authors firstly report crystal chemistry  of ilmentie-type silicate with pyroe composition, whose data quality are high. They also well explain the strong nonlinear behavior of the a- and c-axes with Al2O3 content in the MgSiO3-Mg3Al2Si3O12 system. These results are significant of high-pressure mineral physics, such as in deep earth and shocked meteorite. This manuscript is well organized.  Here I high recommend it in present form. 

Response: Thank you for your review. We have further revised the manuscript based on the others's reviewer comments. I hope the revised manuscript is in a better shape.

Reviewer 3 Report

Comments and Suggestions for Authors

This paper deals with the synthesis and characterization of the structure of a significant earth science compound. The Ilmenite-type compound of composition Mg3Al2Si3O12 was synthesized at 27 GPa in a Kawai-type multi-anvil press at 1073K. Synchrotron X-ray data was collected and cell parameters were compared on the akimotoite-corundum axis. The synthesis of this compound give insight into phases formed at high pressure and temperature upon meteoritic impact.

All parts of the study were well documented, and this paper can be published, yet it could benefit from a small number of additions and/or changes. These additions and/or changes are the following :

-          The overall shape and size of the recovered sample could be included. Given its probable size, an SEM image with an elemental mapping could be considered.

-          Given the electron microprobe analysis composition 115 Mg2.99(2)Al2.01(2)Si3.01(1)O12, it is understandable to set the chemical composition at Mg3Al2Si3O12. However, the assumption that Mg and Si are completely ordered, with Al occupying sites equally, should probably be verified to ensure the correspondence with MgSiO3 in Aki et al paper [7]. If the possibility of such an analysis is possible, it should be included in the paper.

-          a figure with the representation of a well chosen view of each of the five polymorphs in part 4.2 would give the reader a better insight into the differences.

-          the end of the paper could be rounded off by a small but to the point conclusion.

Overall, this paper can be published as is, but would probably gain in interest with a few minor additions.

Author Response

Comment 1: This paper deals with the synthesis and characterization of the structure of a significant earth science compound. The Ilmenite-type compound of composition Mg3Al2Si3O12 was synthesized at 27 GPa in a Kawai-type multi-anvil press at 1073K. Synchrotron X-ray data was collected and cell parameters were compared on the akimotoite-corundum axis. The synthesis of this compound give insight into phases formed at high pressure and temperature upon meteoritic impact.

All parts of the study were well documented, and this paper can be published, yet it could benefit from a small number of additions and/or changes. These additions and/or changes are the following :

Response 1: Thank you for your review. We have made responses to your commenets below.

Comment 2: The overall shape and size of the recovered sample could be included. Given its probable size, an SEM image with an elemental mapping could be considered.

Response 2: Unfortunately, we do not have images and crushed all part of the sample into fine powder finally. Instead, we added descriptions of the sample as below.

Lines 97-98: The recovered sample with a cylinder shape with a diameter of 0.5 mm and a height of 1 mm was identified…..

Lines115-118: SiO2 stishovite with a grain size of ~1 μm was found and included as the secondary phase in the Rietveld analysis. Because the electron microprobe analysis of the recovered Prp-Aki with a grain size of 1-3 μm showed that ……

Comment 3: Given the electron microprobe analysis composition 115 Mg2.99(2)Al2.01(2)Si3.01(1)O12, it is understandable to set the chemical composition at Mg3Al2Si3O12. However, the assumption that Mg and Si are completely ordered, with Al occupying sites equally, should probably be verified to ensure the correspondence with MgSiO3 in Aki et al paper [7]. If the possibility of such an analysis is possible, it should be included in the paper.

Response 3: X-ray scattering factors of Mg, Si, Al are very similar. Therefore, it is almost impossible to examine cite occupancies among these elements. For this reason, we fixed cite occupancies in this manuscript. We added the following sentence to explain this in lines 212-215.

“We note that the present refinement cannot provide information of disordering between the two sites because of similar X-ray scattering factors in these atoms. Future neutron diffraction study may be able to discuss possible disordering.”

Comment 4: a figure with the representation of a well chosen view of each of the five polymorphs in part 4.2 would give the reader a better insight into the differences.

Response 4: We have added a figure to show crystal structures of five polymorphs as Figure 5.

Comment 5: the end of the paper could be rounded off by a small but to the point conclusion.

Response 5: We have added a conclusion section (lines 273-282) as below.

“5. Conclusions

We have synthesized ilmenite-type silicate with Mg3Al2Si3O12 composition at 27 GPa and 1073 K and analyzed its crystal structure by the Rietveld analysis. The refinement showed that volume of the present phase is on the MgSiO3-Al2O3 join whereas lattice parameters are highly anisotropic. Such nonlinear features can be interpreted by changes in oxygen-oxygen bonds with alumina content. Our data clearly showed structural difference between Prp-Aki and Prp-LN phase, both of which have similar but different octahedral arrangements, resulting in a lower density phase of Prp-Aki. The formation of Prp-Aki can be expected in shocked meteorites and make a narrow constraint of shock conditions ranging in 25-27 GPa and 973-1273 K.”

Reviewer 4 Report

Comments and Suggestions for Authors

The authors analyzed the effect of incorporating 25 mol% Al2O3 in the structure of Akimotoite. They determined the unit cell parameters and refined the structure, confirming a non-linear change in unit cell parameters with extent of substitution.

This work was conducted with proper methodology and the discussion is supported by the results

Author Response

Comment: The authors analyzed the effect of incorporating 25 mol% Al2O3 in the structure of Akimotoite. They determined the unit cell parameters and refined the structure, confirming a non-linear change in unit cell parameters with extent of substitution.

This work was conducted with proper methodology and the discussion is supported by the results

Response: Thank you for your review. We have also improved the introduction part based on the other reviewers' comments. We hope the revised manuscript is in a better shape.